# Interconnectedness enhances network resilience of multimodal public transportation systems for *Safe-to-Fail* urban mobility

Zizhen Xu [1] ✉ & Shauhrat S. Chopra [1] ✉

The growing interconnectedness of urban infrastructure networks presents challenges to their ability to handle unforeseen disruptions, particularly in the context of extreme weather events resulting from climate change. Understanding the resilience of interconnected infrastructure systems is imperative to effectively manage such disruptions. This study investigates the role of interconnectedness in enhancing the resilience of public transportation systems in Hong Kong, a city heavily reliant on public transit. Our results demonstrate that interconnected transportation systems improve resilience by reducing topological vulnerabilities, increasing attack tolerance, and enhancing post-disruption interoperability. Findings also identify the potential to integrate vulnerable systems for greater robustness and highlight the marginal benefits of enhancing intermodal transfer. Strengthening interconnectedness among modes of urban public transit fosters a safe-to-fail system, presenting a distinct resilience-by-design approach. This complements conventional resilience-by-intervention approaches that focus on improving individual systems or introducing entirely new systems.

Interconnectedness and interdependencies are ubiquitous in modern infrastructures, which may change their behaviors and characteristics[1] that challenge our past understanding. While no one ignores the criticality of infrastructures and counteractive *fail-safe* measures are established against possible failures, catastrophic failures can still occur. Traditional risk assessment and the *fail-safe* mechanism show limitations when facing unexpected failures. This prompts the proposition of resilience science as a solution with a broader purview than risk[2], alongside the evolution of risk science to incorporate resilience thinking[3]. It is imperative to understand infrastructure resilience, especially when the impact of failures is exacerbated in cities with accelerated reliance on extensive networks of interconnected and interdependent infrastructures.

There is considerable debate surrounding the different approaches to resilience, which can be appropriately classified into two categories: resilience-by-design and resilience-by-intervention[4]. While resilience-by-design focuses more on the system design and immediate response generated within the system (e.g., *safe-to-fail* design research[5]), the resilience-by-intervention approach comprises a wider range of conventional responses that focus on enhancing excess system capacity and galvanizing community support (e.g., intervention and recovery prioritization research[6]). This classification establishes a system boundary for theoretical research and applications that may differ in their emphasis on various aspects of resilience, including different resilience definitions and measures. These approaches to resilience provide complementary implications, and thus, must not be

[1]School of Energy and Environment, City University of Hong Kong, Tat Chee Avenue, Hong Kong SAR, China. ✉e-mail: zizhenxu2@cityu.edu.hk; sschopra@cityu.edu.hk

seen as alternatives. In this paper, we focus on resilience-by-design (henceforth, referred to as resilience) by applying network science to design *safe-to-fail* systems.

Although resilience-by-interventions such as fail-safe engineered measures, and soft infrastructure (such as *deploy, mobilize, and generate*[7]) solutions can mitigate certain disasters, the topology remains the factor that determines the extent of the spread of the disruption. From *fail-safe* to *safe-to-fail*, resilience embraces the inevitability of surprise and develops based on uncertainty and adaptability[5]. A design with a resilience strategy allows for failures, and the system is designed to fail safely without causing catastrophic impacts. Such ability is similar to *graceful degradation* in the resilience literaruture[8]. The goal of resilience is therefore closely linked to sustainability from a long-term perspective, as inter-generation equity of resources and urban sustainability demand more durable infrastructure that fails gracefully during disasters, rather than collapsing and requiring reconstruction.

In previous studies, we have proposed a network resilience framework with three indicators upon the concept of "*safe-to-fail*", including preparedness, robustness, and interoperability[9,10]. These indicators, respectively, are developed to examine whether the network topology is designed well to be ready to handle the failure of any single component, to maintain critical functionality during progressive failures to a significant extent, and to interoperate with the remaining components to temporarily support functionality. This research enhanced the framework and used it to quantify network resilience.

The enhanced framework takes a topological model[11] that focuses on the design of the network topology, which differs from the optimization model[12], data-driven model[13], and probabilistic model[14]. The topology model captures the nature of network-like infrastructures (such as the connection of different locations intrinsic to transportation systems), making it a popular choice for analyzing transportation systems[11,15,16]. It offers an effective top-down approach to modeling interconnected infrastructure and understanding resilience despite the complexity. This research further exploits the capability of the topological model in the resilience assessment from a perspective of safe-to-fail and resilience-by-design.

In the latest decade, resilience-related research is shifting its focus to interconnected and interdependent systems. Interdependency is usually considered a new source of vulnerability in modern infrastructure[17]. Meanwhile, the impact of interconnectedness is dual-faceted. On one hand, the interconnectedness of various critical infrastructures may heighten the likelihood of cascading failures[18]. On the other hand, some infrastructures are pursuing high interconnectedness for better functionality and flexibility, such as transportation, logistics, communication, and supply chain[19,20]. The opposing implications arise from the varying nature of interconnections and the specific entities involved in the interconnection, which might introduce interdependency.

In terms of transportation systems, current research has investigated a wide range of transportation modes[15,16,21,22] but poorly understands how resilience changes when multiple systems are cooperating via interconnection. Many articles have discussed the necessity of future studies on more than one type of transportation systems[23–25]. Also, an intriguing question is whether the past knowledge applicable to a single system remains valid in multimodal systems. Multimodal studies have made progress in model extension and metrics development for topological characteristics[26,27]. Interconnection is regarded as positive by providing complementary service and path options, while the higher or shorter distance in the new paths affects the efficiency of a multimodal system as well as the topological importance of nodes in the network[25]. However, those studies did not incorporate a resilience perspective, and a comparative study that can quantitatively demonstrate the holistic impact of interconnectedness on network resilience is still lacking, leaving unanswered the question of how resilience changes with system interconnectedness and whether there are novel implications arising from multimodal systems when compared to existing knowledge.

Additionally, recognizing the role of geospatial factors in transportation, our study highlights the interconnection modeling in analyzing the multimodal system and its significant geospatial dependencies. In a multilayer network approach, it is not simply an aggregation of multiple networks but hinges on the style of connection between different network layers: the multiplex network where the same set of nodes are connected by different types of edges[23,27] and the interconnected network where different node sets are connected via interlayer edges[26]. To analyze large spatial networks, one can integrate GIS techniques in the modeling that help determine the interrelationship[28]. For public transportation networks (PTN), interconnection is usually defined as the transfer between different modes of transportation. This work argues that interconnections can be beyond real-world structures (such as public transport interchanges and complexes) but depends on the flexible transfer behavior of passengers. In a more general sense, interconnections exist depending on the accessibility of transfer, which can be measured geospatially, such as based on walkability.

Overall, to fill the gaps, this research models a multi-modal public transportation network (MPTN) by integrating each mode of transportation network-by-network and compares the network resilience per state of being isolated and interconnected in each step of integration. Network resilience is quantified using a topology-based resilience framework, predicated on the "safe-to-fail" philosophy. This framework further leverages the potential of topological methods within the realm of resilience. The primary progression of our research is illustrated in Fig. 1. The remainder of the article unfolds with the Results section, which encompasses subsystem characteristics, sequential integration, robustness analysis, and interoperability analysis, followed by the Discussion section. Finally, the Methods section details the metrics used and the implementation of the null model.

## Results

This work demonstrates the approach with a real-world case study of Hong Kong. Hong Kong has significant coverage of public transportation that undertakes about 90% of daily commuting and traveling ridership. The Mass Transit Railway (MTR) and two bus systems (franchised bus and green minibus) are the local mainstays, which are complemented by light rail, ferry, and tram systems. Note that all the six modes are fixed-route public transportations in a "stops and lines" organization manner, where $\mathbb{L}$-space representation is applied for modeling. Other modes that are flexible and ad hoc, such as cycling and taxis, are not included in this study due to their less structured nature which creates inconsistency in the modeling approach and design philosophy. The six subsystems in the MPTN are abbreviated as MTR, LR, FB, GMB, FERRY, and TRAM, respectively in Fig. 2. Characteristics of each mode of PTN are presented in Table 1. PTNs are integrated network-by-network in the analysis, and the changes in the observables are presented in Table 2. Further analysis of resilience-relevant indicators such as network robustness and interoperability are reported in Figs. 3–5.

### Characteristics of subsystems

Taking a general network approach for modeling each mode of the transportation system (namely subsystems) from the Hong Kong case study, we first examined their characteristics before integrating them into an interconnected system. Subsystems are modeled as a group of simple digraphs $\mathcal{G} = \{G_m\}$ where $G_m = (V_m, E_m)$ in $\mathbb{L}$-space representation. As in the graph theory, $V_m$ is the set of nodes (transportation stops) and $E_m$ is the set of ordered edges (transportation links) in a certain mode of transportation $G_m$. The MPTN with all six subsystems consists of a total of 8644 nodes and 15,551 edges.

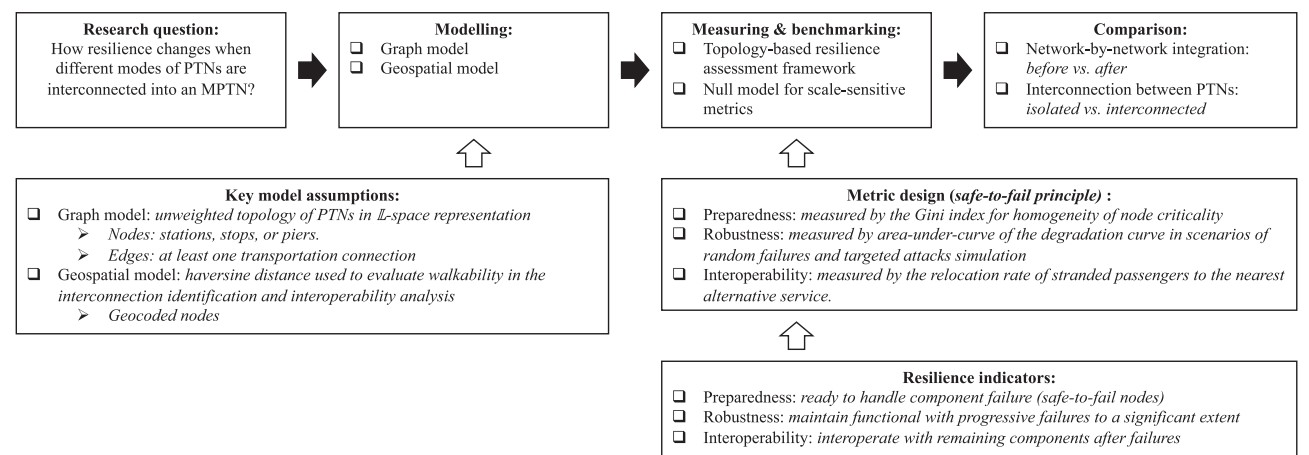

**Fig. 1 | Schematic diagram of the research process.** Comparison of resilience indicators and other characteristics metrics before and after network-by-network integration and per state of being isolated and interconnected lead to the conclusion. In particular, robustness and efficiency metrics are benchmarked with the null model due to its sensitivity to network scale and changing parameters such as average node degree.

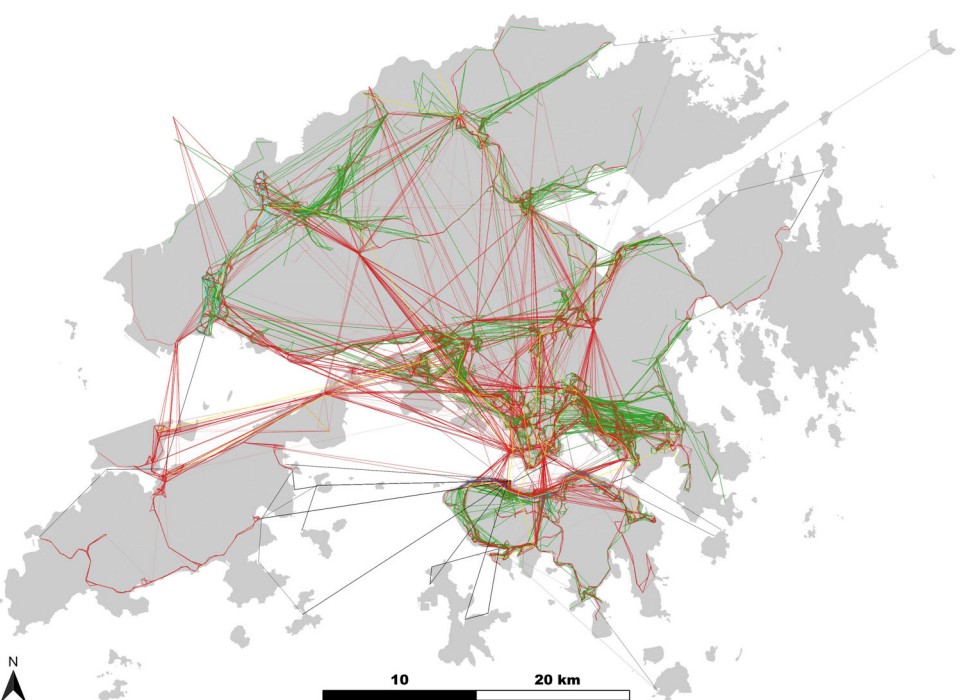

**Fig. 2 | Spatial distribution of the MPTN in Hong Kong.** The six subsystems include MTR (yellow), LR (cyan), FB (red), GMB (green), FERRY (black), and TRAM (blue).

Table 1 lists the characteristics and resilience-related indicators of subsystems in this study. Most metrics have been well defined in the literature, and those relatively less used are introduced in Method. Resilience indicators we developed are based on the resilience definition:[29] *the ability to prepare and plan for, absorb, recover from, and more successfully adapt to adverse events.* For each of the first three in this multiphase definition, we employed an indicator and its topological metric based on the *safe-to-fail* principle: preparedness (Gini), robustness ($r_b$), and interoperability ($Rl$) adapted from the previous work[9]. Adaptation is not discussed in this paper as we consider it a long-term process rather than a measure. The Gini coefficient is used to measure the extent to which sources of vulnerability are distributed across the network, signifying the existence of critical nodes that are not *safe-to-fail*. Network robustness is usually investigated through node percolation[30] (or edge percolation[31]), i.e., removing the nodes sequentially and observing the change of topological properties during network dismantling. We adopt the area under the degradation curve in the percolation as the robustness metric. Transportation interoperability is quantified by a relocation metric that measures the fraction of passengers in a disrupted station that can be relocated to nearby stations/stops, describing a post-disruption response pertaining to immediate recovery efforts (more details in Methods).

Examining the degree distribution (supplementary information), we find that the bus systems, the FB and GMB, show identifiable heavy tails, which is the feature of scale-free networks. It suggests that the bus networks tend to be more robust in the face of random failures but more vulnerable to targeted attacks than random networks. In contrast, the trend is mostly unidentifiable for other subsystems, as the maximum degree of nodes is usually too small to conclude. It is worth mentioning that they have only a handful of nodes with a high degree,

**Table 1 | Some characteristics and resilience-related metrics of the subsystems in this study**

|  | MTR | FB | GMB | LR | FERRY | TRAM |
|---|---|---|---|---|---|---|
| $|V|$ | 96 | 4279 | 4023 | 68 | 61 | 117 |
| $|E|$ | 204 | 9284 | 5689 | 156 | 94 | 124 |
| $\langle k_{out}\rangle$ | 2.13 | 2.17 | 1.41 | 2.29 | 1.54 | 1.06 |
| R | 24 | 1443 | 715 | 22 | 38 | 7 |
| $S_0$ | 1.00 | ≈1.00 | 0.79 | 1.00 | 0.15 | 1.00 |
| $l^{max}$ | 27 | 83 | 107 | 26 | 7 | 84 |
| $\langle l\rangle$ | 10.36 | 14.94 | 23.90 | 9.68 | 0.18 | 32.10 |
| E | 0.15 | 0.09 | 0.03 | 0.18 | 0.04 | 0.06 |
| $E_{geospatial}$ | 0.71 | 0.73 | 0.29 | 0.84 | 0.05 | 0.53 |
| $\langle l_e\rangle$ | 2020 | 1113 | 855 | 440 | 5882 | 245 |
| $\sigma(l_e)$ | 2439 | 2626 | 1658 | 230 | 5342 | 86 |
| Gini (ND) | 0.174 | 0.348 | 0.252 | 0.147 | 0.215 | 0.051 |
| Gini (BC) | 0.490 | 0.611 | 0.757 | 0.456 | 0.727 | 0.188 |
| $r_b$ (Random) | 0.207 | 0.191 | 0.070 | 0.213 | 0.071 | 0.040 |
| $r_b$ (ND-targeted) | 0.080 | 0.075 | 0.021 | 0.090 | 0.040 | 0.024 |
| $r_b$ (BC-targeted) | 0.076 | 0.074 | 0.023 | 0.107 | 0.041 | 0.027 |
| $Rl$ ($d_{max}=750$) | 0.059 | 0.876 | 0.82 | 0.54 | 0.053 | 0.783 |
| $Rl$ ($d_{max}=1600$) | 0.366 | 0.94 | 0.88 | 0.77 | 0.082 | 0.889 |

Note: $|V|$: number of stops; $|E|$: number of links; $\langle k_{out}\rangle$: average out-degree; R: number of directed routes (each for one direction); $S_0$: relative size of largest strongly connected components ($S/|V|$), and $S_0<1$ means the network is not strongly connected; $l^{max}$: maximal shortest path length; $\langle l\rangle$: average shortest path length (path length between disconnected node pair is $l^{max}$); $E$: global efficiency; $E_{geospatial}$: geospatial modification of global efficiency (also known as detour index); $\langle l_e\rangle$: average edge lengths in haversine distance (in meter); $\sigma(l_e)$: standard deviation of edge lengths; Gini (ND) and Gini (BC): two variants of Gini coefficients based on the node degree (ND) and node betweenness centrality (BC), as the preparedness indicator; $r_b$: robustness indicator based on a certain attack strategy (random failure, ND-targeted, and BC-targeted); $Rl$: global average relocation rate with certain relocation distance limit $d_{max}$, as the interoperability indicator.

meaning fewer transportation hubs thereof, which are regarded as sources of vulnerability.

Regarding the node betweenness, interestingly, two rail transport (MTR and LR) and the TRAM system show similar concentrated distribution, i.e., most nodes have medium betweenness centrality. Such a resembling trend may result from a similar design purpose. Rail transportation systems have more distributed shortest paths than bus systems. Less high-centrality nodes mean that the rail transportation system is relatively less likely to trigger a sudden drop in $\langle l\rangle$ in the face of deliberate attacks. On the contrary, the heavy-tail distribution in the GMB and FB networks probably results in more concentrated passenger flows, leading to vulnerabilities to attacks on high-betweenness stations.

Table 1 also shows that the bus transportations (the FB and GMB) have the largest size and number of edges among the five subsystems, and they both have relatively more heterogeneously wired structures than others, as indicated by the Gini value. There are more hubs in the network than others compared to their network size. In contrast, rail transportations (including the MTR, LR, and TRAM) have more homogeneous structures.

In terms of network robustness, the MTR, LR, and FB networks perform better in all three scenarios (supplementary information). They display a high $\langle k_{out}\rangle$, which could be the reason for better robustness. Usually, more edges contribute to the redundancy of the network in terms of connectivity. Examining the degradation curves in a random failure scenario, the MTR and LR networks have similar trends, whereas the LR maintains functionality better than the MTR in the targeted attacks. Moreover, The FERRY, TRAM, and GMB networks display poor robustness in all three scenarios, where a possible reason

**Table 2 | Property of the MPTN in each step of integration with $D_{IMT}$=0 and 100 meters**

| Observables during network integration | Step 1 +MTR | Step 2 +FB | Step 3 +GMB | Step 4 +LR | Step 5 +FERRY | Step 6 +TRAM |
|---|---|---|---|---|---|---|
| $D_{IMT}$=0m (Isolated) | | | | | | |
| $|V|$ | 96 | 4375 | 8398 | 8466 | 8527 | 8644 |
| $|E|$ | 204 | 9488 | 15177 | 15333 | 15427 | 15551 |
| IMT edges | - | - | - | - | - | - |
| $\langle k_{out}\rangle$ | 2.12 | 2.17 | 1.81 | 1.81 | 1.81 | 1.8 |
| $S_0$ | 1.00 | 0.98 | 0.51 | 0.50 | 0.50 | 0.49 |
| $l^{max}$ | 27 | 83 | 107 | 107 | 107 | 107 |
| $\langle l\rangle$ | 10.36 | 18.08 | 70.88 | 71.45 | 71.96 | 72.89 |
| E | 0.15 | 0.08 | 0.03 | 0.03 | 0.03 | 0.03 |
| $E_{geospatial}$ | 0.71 | 0.70 | 0.26 | 0.25 | 0.25 | 0.24 |
| $\langle l_e\rangle$ | 2020 | 1133 | 1029 | 1023 | 1052 | 1046 |
| $\sigma(l_e)$ | 2439 | 2626 | 2315 | 2304 | 2365 | 2356 |
| Gini (ND) | 0.174 | 0.345 | 0.328 | 0.327 | 0.327 | 0.326 |
| Gini (BC) | 0.490 | 0.619 | 0.706 | 0.708 | 0.710 | 0.713 |
| $Rl$ ($d_{max}=750m$) | 0.059 | 0.858 | 0.840 | 0.837 | 0.832 | 0.831 |
| $Rl$ ($d_{max}=1600m$) | 0.366 | 0.927 | 0.905 | 0.904 | 0.898 | 0.898 |
| Z-score ($E$) | 3.154 | 33.021 | 11.428 | 10.092 | 9.942 | 6.430 |
| Z-score ($E_{geospatial}$) | 11.305 | 46.734 | 30.906 | 23.933 | 26.311 | 16.305 |
| $D_{IMT}$=100m (Interconnected) | | | | | | |
| $|V|$ | 96 | 4375 | 8398 | 8466 | 8527 | 8644 |
| $|E|$ | 204 | 9758 | 28771 | 29279 | 29447 | 30699 |
| IMT edges | 0 | 270 | 13594 | 13946 | 14020 | 15148 |
| $\langle k_{in}\rangle$ or $\langle k_{out}\rangle$ | 2.12 | 2.23 | 3.43 | 3.46 | 3.45 | 3.55 |
| $S_0$ | 1.00 | ≈1.00 | 0.99 | 0.99 | 0.99 | 0.99 |
| $l^{max}$ | 27 | 83 | 74 | 74 | 74 | 74 |
| $\langle l\rangle$ | 10.36 | 14.88 | 11.87 | 11.84 | 11.86 | 11.79 |
| E | 0.15 | 0.09 | 0.10 | 0.10 | 0.10 | 0.10 |
| $E_{geospatial}$ | 0.71 | 0.74 | 0.82 | 0.83 | 0.83 | 0.83 |
| $\langle l_e\rangle$ | 2020 | 1103 | 566 | 559 | 575 | 554 |
| $\sigma(l_e)$ | 2439 | 2595 | 1751 | 1737 | 1784 | 1750 |
| Gini (ND) | 0.174 | 0.352 | 0.340 | 0.340 | 0.341 | 0.344 |
| Gini (BC) | 0.490 | 0.611 | 0.706 | 0.707 | 0.707 | 0.708 |
| $Rl$ ($d_{max}=750m$) | 0.059 | 0.878 | 0.931 | 0.931 | 0.929 | 0.930 |
| $Rl$ ($d_{max}=1600m$) | 0.366 | 0.941 | 0.965 | 0.965 | 0.963 | 0.964 |
| Z-score ($E$) | 3.154 | 40.638 | 52.090 | 56.699 | 62.228 | 47.987 |
| Z-score ($E_{geospatial}$) | 11.305 | 66.969 | 52.455 | 68.017 | 73.616 | 50.538 |

can be their low $\langle k_{out}\rangle$. Interestingly, it is observed that the GMB performs worse than the TRAM in the targeted attacks. Considering that the GMB network has a much higher mean node degree than the TRAM, the surprisingly poor robustness of the GMB probably results from its significantly high heterogeneity, which makes the structure easily get fractured by the attacks on critical nodes.

**Change in characteristics during one-by-one integration**

Compared with a single mode of transportation, the MPTN displays different characteristics when interconnected. The interconnection is not always as identifiable as a transportation complex in the real world. Passengers often make transfers between different transportation systems depending on walkability. Therefore, we analyzed the proximity to identify the interconnections and represented them with intermodal edges (edges connecting different modes of transportation in the graph model). The walkable distance can vary among regions, but we do not expect a maximum distance exceeding 1600 meters. The analysis was conducted with the walking distance $D_{IMT}$

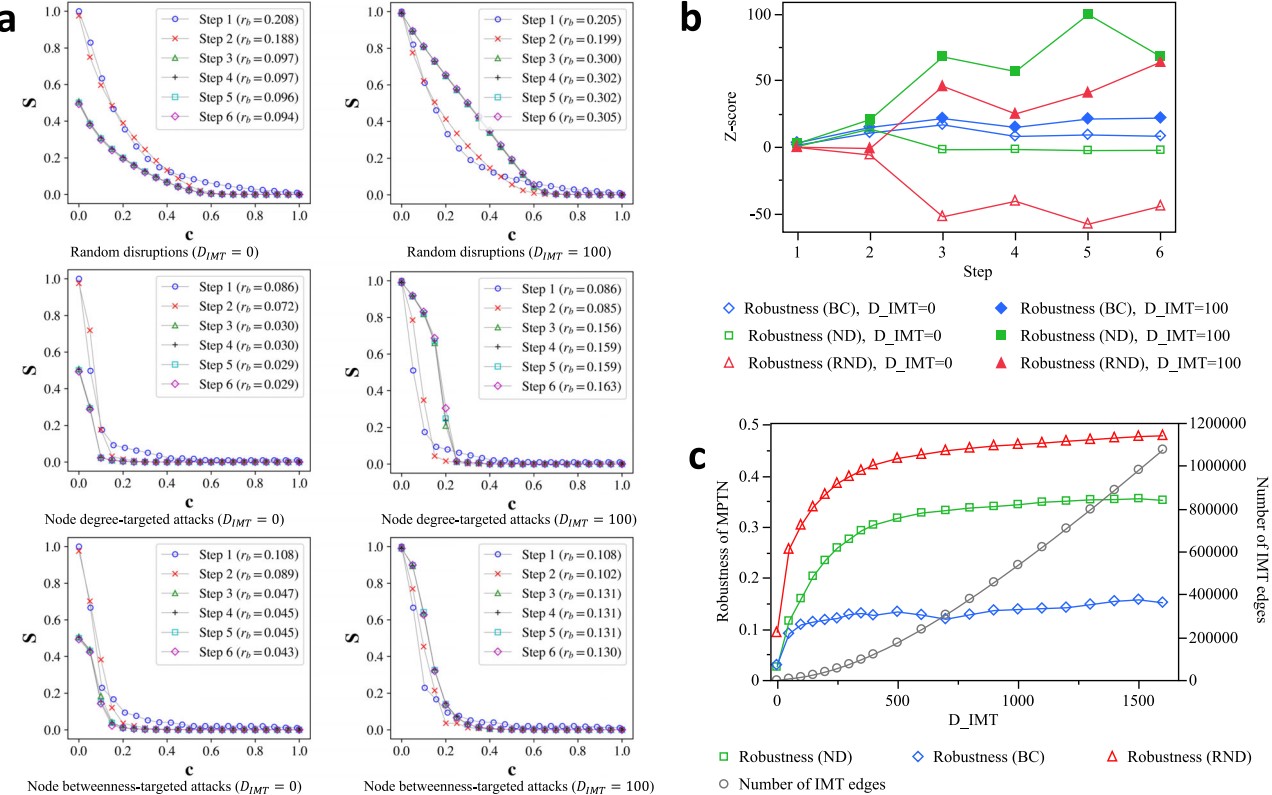

**Fig. 3 | Interconnectedness impact on network robustness. a** Network robustness of isolated (left) and interconnected (right) MPTN at each step of subsystem integration. Each step integrates a corresponding subsystem, i.e., 1: MTR, 2: FB, 3: GMB, 4: LR, 5: FERRY, 6: TRAM. Distance limit for intermodal edges $D_{IMT}$ is set to 0 and 100 meters for comparison. Simultaneously, three failure scenarios are presented: random disruptions (RND), node degree-targeted attacks (ND), and node betweenness-targeted attacks (BC). Interconnected MPTN shows improved network robustness in all three scenarios. **b** The Z-scores of network robustness of MPTN over the null model networks. The null model is used as the benchmark to identify the advantages of interconnected network over random networks. **c** The robustness improvement and the number of IMT edges at different $D_{IMT}$. Marginal improvements in network robustness over increasing IMT distance are different in three disruption scenarios.

varying from 0 to 400 meters in 50-meter steps, and from 400 to 1600 meters in 100-meter steps. Specifically, a 100-meter distance meaning within-2-minute walk was selected to show detailed results, approximating the average size of public transport interchanges (PTIs).

The MPTN is represented as a directed multilayer graph $\mathcal{M} = (\mathcal{G}, E_{IMT})$, which encompasses all the layers $\mathcal{G}$ for different transportation modes and the intermodal edges $E_{IMT}$ that represent transfer links between different modes. We assume intermodal edges to be functionally equivalent to the original edges from a topological perspective, based on our presumption that they offer a level of convenience comparable to "normal transportation links." A distance matrix measuring the haversine distances is computed to determine the pairs of nodes from different subsystems to be interconnected.

To reveal how the MPTN performs differently from individual systems and the impact of integrating each subsystem, we observed the changes in topological properties and resilience indicators of the MPTN during the one-by-one integration of subsystems. Simultaneously, we compared two different $D_{IMT}$ 0 and 100 meters, which means "without" and "with interconnection". The order of subsystems to be integrated is based on their importance in terms of real capacities: MTR, FB, GMB, LR, FERRY, and TRAM.

From Table 2, it is apparent that interconnectedness benefits multiple properties, such as lower $l^{max}$ and $\langle l \rangle$, and higher $E$ and $E_{geospatial}$ (also validated by the Z-scores over the null model). Besides, what is interesting in the data is that $\langle l \rangle$ decreases when high-$\langle l \rangle$ subsystems are integrated (such as GMB and TRAM). It means that multimodal transport provides new paths that are shorter compared

to monomodal transport. Taken together, these results provide evidence that there are topological advantages of interconnected MPTN.

Additionally, Table 2 shows that the Gini (ND) of the interconnected system is slightly higher than that without interconnection (by 5.5%). The MPTN grows to have a slightly more heterogeneous connection pattern during the integration of subsystems. A possible explanation might be the tendency to add intermodal edges to the high-degree nodes, while the low-degree nodes are not interconnected proportionally. Moreover, one unanticipated finding is that the Gini (BC) of the interconnected system is roughly the same as the one without interconnection (0.7% decrease). It indicates that, in this case study, interconnectedness does not significantly affect the distribution of node betweenness centrality. However, it is difficult to explain these results due to the complexity of analyzing how all-pair shortest paths are changing, and future work may investigate the reasons behind this.

### Interconnectedness and network robustness of MPTN

Figure 3 compares the network robustness of isolated and interconnected MPTN at each step of subsystem integration in three failure scenarios. The results display that interconnectedness significantly benefits robustness. Specifically, Fig. 3a shows degradation curves, where there is a prominent change of curve shape in the interconnected system (targeted attacks, $D_{IMT} = 100$). At the beginning of percolation, the interconnected system exhibits high tolerance to attacks (especially on the node degree). In other words, the MPTN shows improved robustness in the face of disruptions in

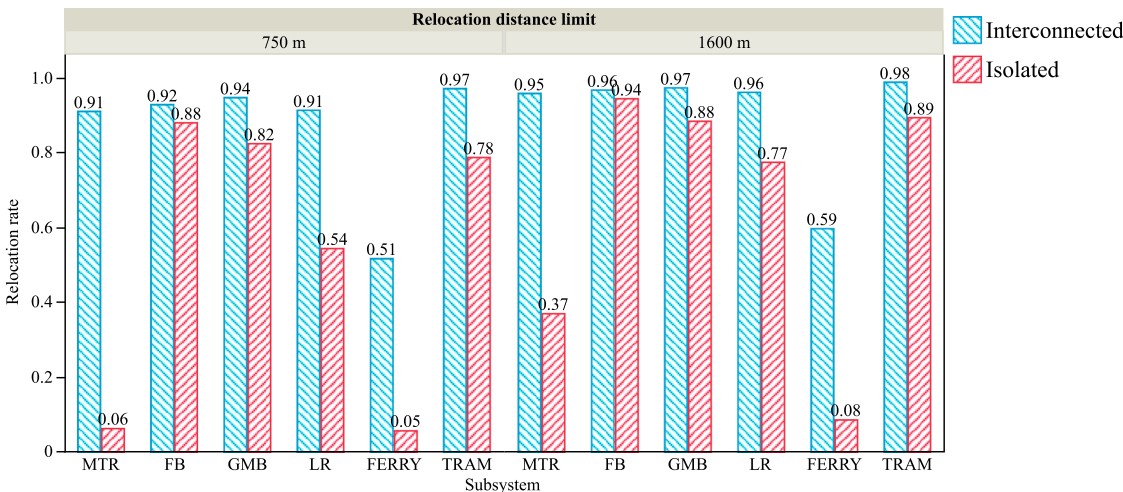

**Fig. 4 | Interconnectedness impact on relocation capability of subsystems.** Relocation rates of subsystems when isolated (red) and interconnected (blue) are presented with 750 and 1600 meters relocation distance limit. MTR, LR, and FERRY networks show significant improvements when interconnected.

transportation hubs compared to individual PTNs. This finding can help planners in developing solutions for PTNs that show that type of vulnerability.

Also, considering the possible effect of different network sizes and $\langle k_{out}\rangle$, a null model is developed to validate the results. The null model integrates the Erdős–Rényi model with geospatial constraints to capture the tendency of edge lengths in real-world transportation systems. $Z$-score is employed to indicate the degree to which the indicator deviates from the average of null mode networks. Figure 3b presents the $Z$-scores of robustness results, where the MPTN outperforms the isolated competitor in all three scenarios. Furthermore, it is worth noting that the improvement in network robustness (BC-targeted attacks) is relatively small. It may be explained by Gini (BC), which indicates that the node betweenness inequality is roughly unchanged.

Another important finding from Fig. 3 is that integrating a vulnerable network through interconnection may gain significant improvement in robustness, which is observed when integrating GMB and TRAM (step 3 and 6). The reasons can be that the one to be integrated has strong interconnections with the existing MPTN (this can be confirmed by the number of IMT edges), or it reduces the topological vulnerability of the existing MPTN (probably due to the complementary nature of rail and bus transport). Similarly, integrating a robustness network may be counterproductive, which is observed in step 4 (LR), and it may be due to the fact that the LR network only serves communities rather than the whole city.

Furthermore, the relationship between the robustness of the MPTN and the maximum distance of intermodal transfer is presented in Fig. 3c. As the willingness to walk for intermodal transfer (IMT) varies among passengers in different regions, a range of $D_{IMT}$ is considered from 0 to 1600 meters. It is clear that the robustness of MPTN increases with $D_{IMT}$, where different upper limits are observed in three scenarios. Robustness (BC) reaches the limit at the earliest, followed by robustness (ND). Surprisingly, the robustness (RND) approaches the theoretical limit (0.5 for a *complete graph*) at a 1600-meter IMT distance, which is a smaller distance than the authors' anticipation. Also, we find that the marginal benefit of robustness improvement is significant at a short IMT distance, where the costs (the number of IMT edges needed) are relatively small. The implication is that enhancing the transfer is a promising way to improve network robustness, especially when the current systems are poorly interconnected. Moreover, the robustness for degree-based attacks and betweenness-based attacks is at the same level at the beginning (when without interconnection), but the former gets more improvement from the interconnectedness than the latter.

## Network interoperability of MPTN

In terms of network interoperability, proximity-based relocation of unweighted networks has been previously applied to London metro[32] and its weighted variant has been developed to compare five metro systems[10] based on the shortest paths of Origin-Destination flows. This research improved the unweighted method to include the connection of OD pairs before and after a disruption (i.e., reachability).

Transportation interoperability of individual subsystems and the MPTN can be compared in Fig. 4. All subsystems show improved relocation capability when interconnected, while the MTR, LR, and FERRY get significant benefits. The global average relocation rate of MTR increases from 0.06 (very poor) to 0.91 (very good) as a part of interconnected MPTN at a 750-meter relocation distance limit. Similarly, FERRY shows great improvements when the relocation distance limit is either 750 or 1600 meters. It means that other subsystems are vitally important to the MTR and FERRY post-disruption. In other words, they depend on other subsystems in the post-disruption scenario to redistribute the passengers. In contrast, FB, GMB, and LR get minor improvements due to their densely placed stops. An overall relocation rate of 0.93 is observed in the completely integrated MPTN (relocation distance limit = 750 m). This number indicates significant interoperability of the MPTN, which represents that, in single disruption scenarios, approximately 93% of passengers can continue their trips after relocations.

For a spatial representation of interoperability, we employ the area interoperability metric to display our findings. As an example, we have visually depicted the area interoperability of MPTN for subdistricts in Hong Kong, officially *small tertiary planning unit groups* in Fig. 5. This metric can be interpreted as the number of fully interoperable stations per kilometer square. The subdistricts near the HKU MTR station (No. 116), the northeast part of Yau Ma Tei (No. 220), Wan Chai South (No. 131 and 132), the border of Central and Sheung Wan (No. 114), Mong Kok East (No. 222), the west part of To Kwa Wan (No. 241), and Kowloon City (No. 285) have the top area interoperability, indicating that there are more stations that have a high relocation rate within the area.

## Discussion

Generally, this work investigates the topological impact of interconnecting different transport modes within the public transportation sector using a real-world case study from Hong Kong. Building *safe-to-fail* mobility in cities requires a transportation system designed with components that can fail safely, a robust topology that maintains connectivity during failures, and interoperability that enables immediate redistribution of passengers. This research identifies the

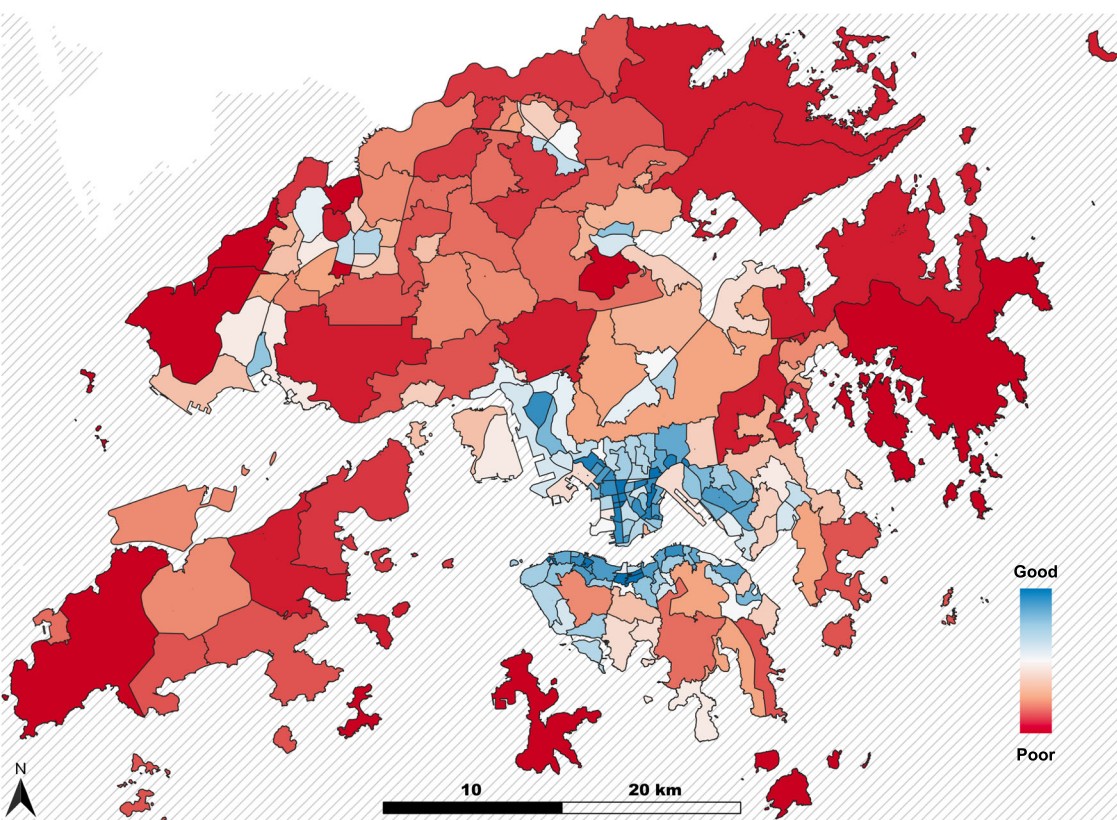

**Fig. 5 | Area interoperability of subdistricts (*small tertiary planning unit groups*) in Hong Kong, colored based on their ranking from good (blue) to poor (red).** The list of subdistricts is available in the dataset, titled 2016 Population By-census Statistics (By small Tertiary Planning Unit Group), in the Hong Kong Geodata Store (geodata.gov.hk).

positive impacts of interconnectedness on the network robustness, interoperability, and efficiency in a multimodal public transportation network (MPTN). The MPTN in Hong Kong exhibits strong tolerance to minor disruptions after integrating three subsystems: MTR, light rail, and buses, and shows increased robustness when facing both random disruptions and attacks on interchange stations and hubs. The significant potential for beneficial interconnection between rail and bus transit systems that we have demonstrated in our research may offer important insights for regions looking to enhance the resilience of their public transportation systems.

Meanwhile, our research finds a slight centralization of the network structure from an equity point of view, potentially resulting in greater difficulty and cost in implementing preparedness measures before a disruption. A centralized system structure, characterized by a handful of exceedingly interconnected interchange stations, does not align with the "safe-to-fail" principle. To mitigate this vulnerability, it is recommended to focus investments on developing satellite stations rather than central ones. These findings also suggest that trade-offs exist between network preparedness, robustness, and efficiency. Achieving a balance among them, and even more conflicting factors such as cost, carbon, land use, and political will, requires multicriteria analysis when making decisions in real-world applications due to the complex and multifaceted nature of resilience.

Bridging the theory and action, planners can take into account resilience indicators in the stage of multi-criteria decision-making. This can be a good approach to incorporate resilience-by-design thinking. All the proposed resilience indicators can be adapted and improved for practical use, such as creating weighted variants, depending on the interest and requirements of stakeholders. While certain indicators may prove challenging to interpret and achieve a shared understanding within the operational context, this issue can be partially

addressed through comprehensive benchmarking and comparative studies of real-world transportation systems[33].

Our analysis also provides a distinct insight that interconnecting vulnerable systems can improve the overall network robustness. In the field of power supply, there are already similar examples of exploring integrating microgrids into large power systems for resilience enhancement[34]. Intermodal edges provide an opportunity to adjust the network topology and address the source of vulnerability. In particular, our results from Hong Kong demonstrate that interconnectedness benefits network robustness more in the face of attacks on hubs and transfer stations than on high-betweenness stations, with the former exhibiting more than twice the improvement. These findings suggest a potential solution to the common vulnerability of public transportation networks at their hubs and transfer stations. Transportation planners from other regions could explore enhancing interconnection to address similar issues.

The examination of robustness at different intermodal transfer (IMT) distances reveals a concave-down relationship, where the marginal benefit of robustness improvement peaks when the IMT distance is short. It suggests that there can be considerable benefits to enhancing intermodal transfer in a transportation system with poor IMT accessibility, by balancing the improvement in robustness with the action cost. Examples of such actions include constructing transfer passageways and footbridges, and setting up interchange sites to improve walkability. However, these actions require collaborative efforts between different transportation operators and city planners. Future work may explore the influence of pedestrian on the resilience of public transportation systems and its regional variation.

Furthermore, the Hong Kong MPTN exhibits significant interoperability in single disruption scenarios due to its densely placed stations. Such interoperability facilitates a *safe-to-fail* system design by

**Table 3 | Databases of the fixed-route transportation system in Hong Kong are available in the data portal (https://data.gov.hk/en/). Some databases follow the GTFS format, which stands for General Transit Feed Specification**

| PTN | Mode | Data access year | Database provider and name | PTN description |
|---|---|---|---|---|
| MTR | Rail transport | 2022 | MTR Corporation Limited: *MTR routes, fares, and barrier-free facilities* | Local metro system, namely *Mass Transit Railway*. |
| LR | Rail transport | 2022 | MTR Corporation Limited: *MTR routes, fares, and barrier-free facilities* | Light rail system. |
| FB | Road transport | 2022 | Transport Department: *Headway information of public transport services* | Franchised bus services consisting of multiple fixed-route bus systems by different operators, including *KMB, CTB, NWFB, CTB, DB, LWB, LRTFeeder, NLB, PI, and XB*. |
| GMB | Road transport | 2020 | Transport Department: *Routes and fares of public transport* | Public green minibus service operated with fixed-route 16 or 19-seat minibusses. |
| FERRY | Water transport | 2022 | Transport Department: *Headway information of public transport services* | Ferry service connecting public piers. |
| TRAM | Road transport (track) | 2022 | Transport Department: *Headway information of public transport services* | Vintage tram service. |

redistributing stranded flows. This study provides a good example that local MTR and FERRY systems have limited capacity to relocate stranded passengers on their own, and interconnecting with other transportation modes provides an effective solution to this problem. It is worth noting that the MTR company already owns some bus routes of the franchised bus network (LRTFeeders in Table 3) for daily operations and contingency relocation. This solution could be applicable to other transportation systems with similar weaknesses, and merits consideration by transportation planners.

In conclusion, our findings suggest that interconnectedness can offer a distinct approach to enhancing transportation resilience, beyond simply improving each system in isolation or introducing entirely new systems. Operators and planners conventionally utilize established strategies like contingency planning to address journey reliability and disruption risks, reflecting a resilience-by-intervention approach, while network science offers a complementary resilience-by-design perspective that emphasizes the system's topology. Also, we believe that the implications drawn from our generalizable network approach may be applicable to other network-like infrastructures seeking to develop *safe-to-fail* designs. The primary dataset utilized in this study adheres to the General Transit Feed Specification (GTFS) format. This choice facilitates generalizability, validation, and replication of the analysis for future research. There is a need for transport departments to use and create standardized metrics for standardized data collection approaches like GTFS.

This paper has highlighted the topological advantages of interconnectedness in the public transportation sector, and future work can expand on this analysis to consider weighted systems in terms of demands, capacity, and other features[35] in order to provide more operational solutions. We look forward to future research delving into the effect of varying weighing methodologies and real-world regional discrepancies. Moreover, it is important to note that inter-sectoral connections, such as those between the transportation and energy sectors, present a distinct scenario, particularly when interdependencies are involved. We must exercise caution when applying these findings to real-world contexts due to the potential for emergence beyond what can be captured by modeling and theory.

## Methods
### Network preparedness
From a topological perspective, structural vulnerability arises from node inequality. The presence of critical nodes contributes to the network's vulnerability, thus we regard these nodes as sources of vulnerability. The network is not prepared to handle failures related to these critical nodes if no additional protective measures are in place. In this research, network preparedness is gauged by analyzing how the vulnerability sources are distributed across the network from a homogeneity perspective. It indicates the extent to which the topology is designed to minimize the consequence of all possible node failures. In other words, every node in a prepared network is *safe to fail* itself. Gini coefficients of the node degree metric and node betweenness centrality metric are computed as the network preparedness indicator:

$$Gini(M) = \frac{\sum_{i=1}^{N}(2i - N - 1)M_i}{N\sum_{i=1}^{N}M_i} \tag{1}$$

where $M_i$ is the $i$-th smallest value of node degree or betweenness; $N$ is the number of nodes in the network. Gini values vary from 0 to 1, indicating the most homogeneous and heterogeneous case (such as the lattice and star network). Note that what the Gini index captures varies upon the metric selected to represent the node criticality, e.g., node degree and node betweenness centrality, which in turn limits the scope of preparedness to be topological. More choices of node metrics can expand the scope of preparedness measurement, such as

closeness centrality, flows in a node, and other indicators associated with real-world operation.

## Network robustness

Network robustness is examined through node percolation processes. We continuously evaluate the relative size of the largest strongly connected components (denoted as $S$) in the network to describe the degradation of the network structure. Besides, node removal can be random or targeted (following certain prioritization strategies). Two widely used strategies are prioritizing nodes by the node degree and node betweenness centrality. In the process of targeted attacks, we re-identify the highest node degree and centrality after every step of node removal, which usually changes with network dismantling. Also, there might be multiple nodes with the same value of node degree or centrality, and we further apply random choice. Considering the computational cost, we repeat the node percolation 1000 times for random disruptions and 100 times for node degree-targeted and centrality-targeted attacks (where the same centrality value is far less possible) to get reproducible results. Additionally, the computation of betweenness centrality after every node removal could be expensive; we thus remove nodes at step-by-step 5%. In each step, we identify the nodes with the highest node degree or node betweenness centrality recalculated based on the damaged network topology and remove them from the network. Because multiple removals are being used, we employ the integral-of-curve method instead of the average-value method[36].

$$r_b = \int_0^1 S(c)dc \quad (2)$$

where $c$ is the fraction of nodes removed and $S$ is the size of the largest strongly connected component normalized to the number of nodes in the unperturbed network. This research employed a widely recognized performance index $S$ and specifically tailored it (to be strongly connected) for the analysis of the directed graph model. This assumption can be adjusted depending on varying perspectives regarding network connectivity and structural integrity.

## Network efficiency

Efficiency metrics are usually employed to indicate the functionality of networks. However, the traditional efficiency formula is not sufficient in analyzing a real-world transportation network subject to certain physical limitations in the network design. In comparison with random networks, real-world transportation networks can have lower efficiency, and hence an additional geospatial modification is applied to indicate the efficiency over the spatial limitation[37].

$$E_{geospatial} = \frac{1}{N(N-1)} \sum_{i \neq j \in V} \frac{d(i,j)}{\ell(i,j)} \quad (3)$$

where $N$ is the number of nodes in the network; $d$ is the haversine distance between nodes and $\ell$ is the haversine distance along the shortest path. The geospatial efficiency metric compares the flight distance and the travel distance for all node pairs, enabling us to assess the efficiency of paths in terms of their geospatial limit. A maximum ratio of 1 indicates that a path is the geospatially shortest between two nodes. This study employed the haversine distance as a simplified approach for measuring distances. However, it is worth noting that further improvements could be made by incorporating data that provides actual travel distances if necessitated by specific research questions and real-world applications.

## Transportation interoperability

Transportation interoperability is quantified by a metric of node relocation rate computed through the estimation of proximity and reachability (Eqs. 4, 5), which we understand as a short-term recovery

(different from the restoration of disrupted components). We assume a single disruption of a node $v$ that has multiple neighbors within a distance of $d_{max}$. For every previous reachable node, we search for the nearest neighbor where we can find a new path and compute the linear distance decaying factor $df(l)$.

$$\text{Node relocation } Rl(v) = \frac{1}{N} \sum_n df(\min l_{v,u|n}) \quad (4)$$

$$df(l) = 1 - \frac{l}{d_{max}}, \quad 0 < l \leq d_{max} \quad (5)$$

where $N$ is the number of reachable nodes from node $v$ in the unperturbed network; $n$ is any of the reachable nodes, and $l_{v,u|n}$ is the distance to the neighboring node $u$, from which we can find a new path to node $n$ when node $v$ is disrupted. $d_{max}$ is the relocation distance limit. This research took 750 meters and 1600 meters in the analysis (according to the survey[38] and the previous work[32]). Please note that for distance measurement, we have utilized the haversine distance and incorporated a linear decaying factor as an assumption to simplify the model. It is important to acknowledge that these simplifications can be further refined with the availability of more precise data that support the modeling of pedestrian networks, which can provide accurate walkability analysis during adverse events such as flooding[39].

In addition to the node relocation, we compute the network-level relocation indicator $Rl$ by the global average of all nodes in the network (Eq. 6). Besides, an area interoperability indicator (Eq. 7) for each planning group in Hong Kong is computed to visualize spatial differences.

$$\text{Network relocation } Rl = \frac{\sum Rl(v)}{N} \quad (6)$$

$$\text{Area interoperability} = \frac{\sum Rl(v)}{Area} \quad (7)$$

## Null model benchmark

Networks with different network sizes and numbers of edges may exhibit varied characteristics by nature. Some properties are especially sensitive, such as network robustness. To find out whether there are advantages of MPTN in terms of its robustness in comparison to random networks, we introduced a null model as the benchmark. In theoretical comparison, the Erdős–Rényi (ER) model is a primary choice[40]. However, real-world transportation systems show a certain connection trend that the ER model cannot capture. For example, metro and bus systems usually control the real distance between stops, while the ER model tends to have more long edges ignoring the geospatial constraints. It leads to significantly higher global efficiency in ER model compared with real-world transportation systems. Therefore, we integrated the ER model with the geospatial constraints.

Compared to the original network, the proposed null model generates networks with the same node set and a number of edges, similar average edge lengths (in haversine distance), and their standard deviation. Based on the $G(n, p)$ model of ER, we rewire the network by following the discrete distribution of edge lengths in the original network. Specifically, each random number is generated upon the discrete distribution of edge lengths, and the network includes the edge with a length closest to that number from all $N \times (N-1)/2$ potential edges. In this way, the null model produces approximately the same edge length distribution, which roughly reflects the distance-related cost. To benchmark the indicators in need, $Z$-score[40] is calculated based on the average $\bar{x}_N$ and standard deviation $\sigma_N$ from a group

of null model networks.

$$Z - score = \frac{x - \bar{x_N}}{\sigma_N} \qquad (8)$$

## Statistics and Reproducibility

To ensure reproducibility, node percolation collected results from random simulations with 1000 repetitions for random disruptions and 100 repetitions for targeted attacks. Null model benchmark collected results from 50 simulations. To balance precision and computational cost, we primarily controlled the Standard Error of the Mean to below 0.01. No data were excluded from the analyses.

## Data availability

Raw data are available in the online data portal, and Table 3 lists the database name and provider. The raw data used in this study have also been deposited in Zenodo under accession code https://zenodo.org/badge/latestdoi/657060246. The results data are provided in the Source Data file. Producing Fig. 5 involved additional data from 2016 Population By-census Statistics (By Small Tertiary Planning Unit Group) published in Hong Kong Geodata Store. Source data are provided with this paper.

## Code availability

Graph modeling and topological analysis are performed mostly using *NetworkX* (https://networkx.org). Other relevant codes[41] are available in https://zenodo.org/badge/latestdoi/657060246.

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

## Acknowledgements

The work described in this paper was supported by the General Research Fund (GRF) (Project No. CityU 9048170, S.S.C.) and the Collaborative Research Fund (CRF) (Project No. C1105-20G, S.S.C) grants from the Research Grants Council of the Hong Kong Special Administrative Region, China. In addition, we appreciate the long-term internal support for research on resilient infrastructure systems from the City University of Hong Kong.

## Author contributions

Z.X. collected data and performed analyses. Z.X. and S.S.C. designed the research and contributed to paper preparation.

## Competing interests

The authors declare no competing interests.
