## [Peer Review File · Nature Communications]

Interconnectedness enhances network resilience of multimodal public transportation systems for *Safe-to-Fail* urban mobilityREVIEWER COMMENTS

Reviewer #1 (Remarks to the Author):

The article represents an advance in our understanding of the resilience of (multimodal) transportation networks (and by extension, of networked lifelines) and hence, in my opinion, should proceed towards publication. However, I feel there are a few relatively minor issues that need to be addressed prior to publication.

I would have liked to see a clearer and sharper delineation of the design of experiment for the core hypothesis (i.e., as expressed in the current title) along with the metrics used to establish or falsify the hypothesis. I think the basics are already there, but being a little more explicit, especially in the falsifiability, may help justify the conclusions better.

A discussion of the metrics used for, e.g., preparedness, efficiency, robustness, and interoperability, in terms of their applicability (and caveats and assumptions may be useful to better appreciate the validity of the conclusions and where more information could be useful for future research. A discussion on the assumptions and the value of data gathering could be useful.

One criticism, not of this work specifically but of network science and graphical methods in lifelines including transportation, has been the inability to transition to actionable insights. A clearer discussion on what interventions or design principles, and perhaps investments strategies or policy principles, the insights and methods here can inform could be useful in that regard.

Overall, I find the problem well formulated, the societal motivation and the literature well presented, and the conclusions sound. I would recommend publication after the authors get a chance to respond to the comments above.

Reviewer #2 (Remarks to the Author):

This paper takes a topology-based network approach to assess multiple network resilience indicators and found that the connections between subsystems can actually enhance the resilience of the transportation network instead of making it more fragile.

The paper is mostly of good quality with some areas needing improvement.

Structure:

This paper's construction creates a lot of confusion.

Results section comes before Methods section. It's confusing to read this at first. When I read the Results section, I had a few queries, and the Methods section answered most of them.

Besides this, some paragraphs should be moved. For example, lines 230-233 are suggested to be moved to the section where preparedness is discussed (lines 152-153, so it is clear why Gini values are used as preparedness indicator. Similarly, lines 246-249 are suggested to be moved to the section where robustness is discussed, and lines 299-301 to be moved to the location where response and recovery measures are described.

Methods:

This paper takes edge length as the only factor when performing betweenness centrality analysis. This may be applicable to train and subway systems, as the speed of these forms of transportation may be comparable. Nevertheless, the speed of buses may vary due to the speed limit and traffic congestion. Thus, in the instance of a transportation network, the process of determining the shortest path, which is vital for centrality calculation, could be more precise if 'shortest time' was utilized over 'shortest distance'.

Is the actual route distance data available, so that the authors can replace the haversine distance

with the actual route distance?

Other comments:

1. This paper used different terms, such as 'intermodal' (line 21) and 'multimodal' in most other places. Unifying the terms is suggested to avoid any misunderstanding.
2. Line 58: 'is fail' => 'fails'
3. Line 137: Please specify that the number of nodes and edges are for the interconnected system. It isn't easy to understand from a first glance, as the prior sentence explains the subsystem.
4. Caption missing from Table 1.
5. I can understand that interoperability is an indicator of 'response'. But how it can represent 'recovery' is not clear since relocation does not restore the functionality of the system.
6. Line 181, should be Table 1?
7. Line 234, should be Table 2?
8. Line 275, should be Fig. 2?
9. Line 364, extra ')' at the end.

In summary, I recommend major revision before this paper is accepted.

Reviewer #3 (Remarks to the Author):

This paper describes results before the methods - very difficult to follow.

The concepts of "integration" (which seems to be much of the focus of the analysis) and "safe-to-fail" (which seems to be much of the focus of the title and introduction section) need to be better aligned. Quite a lot of the introduction material is restatement of resilience literature found in many, many sources. It's recommended that the authors stick to what's new about this paper and how it fits in the landscape of resilience, not just providing a bunch of terms dealing with resilience.

It is unclear how the Gini index measures network preparedness.

Both Mass Transit Railway and metro are abbreviated with MTR.

Response to comments

We sincerely appreciate the time and effort the reviewers have dedicated to reviewing our work. The input has undoubtedly enriched our work, and we believe the enhancements made in response to the comments have greatly improved the quality and clarity of the paper. A revised manuscript is provided with track-change (following the clean version).

REVIEWER COMMENTS #1

-	The article represents an advance in our understanding of the resilience of (multimodal) transportation networks (and by extension, of networked lifelines) and hence, in my opinion, should proceed towards publication. However, I feel there are a few relatively minor issues that need to be addressed prior to publication.	We appreciate the thoroughness with which the reviewer reviewed the work and provided constructive feedback.
1	I would have liked to see a clearer and sharper delineation of the design of experiment for the core hypothesis (i.e., as expressed in the current title) along with the metrics used to establish or falsify the hypothesis. I think the basics are already there, but being a little more explicit, especially in the falsifiability, may help justify the conclusions better.	We agree with the reviewer and have included a flowchart (the new Figure 1) with experiment design, metrics, and assumptions used to establish or potentially falsify the hypothesis. Following figure has been added to the manuscript. (Big figure in the revised manuscript)  Figure 1 Schematic diagram of the research process. Comparison of resilience indicators and other characteristics metrics before and after network-by-network integration and per state of being isolated and interconnected lead to the conclusion. In particular, robustness and efficiency metrics are benchmarked with null model due to its sensitivity to network scale and changing parameters such as average node degree.
2	A discussion of the metrics used for, e.g., preparedness, efficiency, robustness, and interoperability, in terms of their applicability (and caveats and assumptions may be useful to better appreciate the validity of the conclusions and where more information could be useful for future research. A discussion on the assumptions and the value of data gathering could be useful.	We have added texts about assumptions and applicability of metrics in Methods section as well as the value of data gathering in Discussion. Following texts have been added to the manuscript. Methods > Network preparedness (line 448 in the clean version): Gini values vary from 0 to 1, indicating the most homogeneous and heterogeneous case (such as the lattice and star network). Note that the Gini index varies depending upon the network metric selected for node criticality, e.g., node degree and node betweenness centrality, which in turn limits the scope of preparedness to be topological. Other similar metrics can expand the scope of preparedness measurement, such as closeness centrality, flows in a node, and other indicators associated with real-world operation. Methods > Network robustness (line 473): This research employed a widely recognized performance index S and specifically tailored it (to be strongly connected) for the analysis of directed graph model. This assumption can be adjusted depending on varying perspectives regarding network connectivity and structural integrity.

		Methods > Network efficiency (line 486): The geospatial efficiency metric compares the direct distance and the along-the-path distance for all node pairs, enabling us to assess the efficiency of paths in terms of their geospatial limit. A maximum ratio of 1 indicates that a path is the geospatially shortest between two nodes. This study employed the haversine distance as a simplified approach for measuring distances. However, it is worth noting that further improvements could be made by incorporating data that provides actual travel distance if necessitated by specific research question and real-world applications. Methods > Transportation interoperability (line 505): Please note that for distance measurement, we have utilized the haversine distance and incorporated a linear decaying factor as an assumption to simplify the model. It is important to acknowledge that these simplifications can be further refined with the availability of more precise data that support modeling of pedestrian network, which can provide accurate walkability analysis during adverse events such as flooding³⁹. Discussion (line 422): The primary dataset utilized in this study adheres to the General Transit Feed Specification (GTFS) format. This choice facilitates generalizability, validation, and replication of the analysis for in future research. There is a need for transport departments to use and create standardized metrics for standardized data collection approaches like GTFS.
3	One criticism, not of this work specifically but of network science and graphical methods in lifelines including transportation, has been the inability to transition to actionable insights. A clearer discussion on what interventions or design principles, and perhaps investments strategies or policy principles, the insights and methods here can inform could be useful in that regard. Overall, I find the problem well formulated, the societal motivation and the literature well presented, and the conclusions sound. I would recommend publication after the authors get a chance to respond to the comments above.	We agree with the reviewer. The topological approach has its limitation, and gaps exist between reality and the models. Conventionally, operators and planners utilize established strategies to address issues such as journey reliability and disruption risks, with contingency planning being a common solution, which reflects a resilience-by-intervention approach. On the other hand, network science places emphasis on the system's topology, representing a resilience-by-design perspective. These two perspectives are complementary in the planning and operation. Bridging the theory and action, planners can take into account resilience indicators in the stage of multi-criteria decision-making. This can be good approach to incorporate resilience-by-design thinking. All the resilience indicators can be adapted and improved for practical use, such as creating weighted variants, depending on the interest and requirement of stakeholders. Certain indicators may prove challenging to interpret and achieve a shared understanding within the operational context. However, this issue can be partially addressed through comprehensive benchmarking and comparative studies of real-world transportation systems[R1]. In terms of specific principles/strategies suggested by our findings, we list some as follows:  ▪ When considering new investments, it is beneficial for resilience of entire transportation sector to factor in the walkable distance between planned and

		existing stations to promote interchange. However, this approach may present a certain degree of conflict with profitability objectives for transportation operators.  ▪ Through collaborative efforts between transportation operators and city planners, the establishment of intermodal connections is an effective strategy for resilience. This approach offers considerable marginal benefits, particularly when such connections have not been implemented before. ▪ A centralized system structure, characterized by a handful of exceedingly interconnected interchange stations, does not align with the "safe-to-fail" principle. To mitigate this vulnerability, it is recommended to focus investments on developing satellite stations rather than central ones. ▪ Although interconnectedness is advantageous within the public transportation sector, it's important to note that inter-sectoral connections, such as those between the transportation and energy sectors, present a distinct scenario, particularly when interdependencies are involved. [R1] I. Linkov and B. D. Trump, The science and practice of resilience. Springer, 2019. Above texts have been incorporated into different paragraphs in the Discussion section (see the track-change manuscript).
--	--	--

REVIEWER COMMENTS #2

-	This paper takes a topology-based network approach to assess multiple network resilience indicators and found that the connections between subsystems can actually enhance the resilience of the transportation network instead of making it more fragile. The paper is mostly of good quality with some areas needing improvement.	We greatly value the feedback and have made the necessary revisions to the text in response.
1	Structure: This paper's construction creates a lot of confusion. Results section comes before Methods section. It's confusing to read this at first. When I read the Results section, I had a few queries, and the Methods section answered most of them.	We have added brief methodology descriptions and a flowchart (new Figure 1) to the Introduction section to improve the readability. Please note that we keep the section order (introduction, results, discussion, methods) as required by the journal. Following texts have been added to the manuscript. Introduction (line 62 in the clean version): In previous studies, we have proposed a network resilience framework with three indicators upon the concept of "safe-to-fail", including preparedness, robustness, and interoperability^{8,9}. These indicators, respectively, are developed to examine whether the network topology is designed well to be ready to handle failure of any single

component, to maintain critical functionality during progressive failures to a significant extent, and to interoperate with the remaining components to temporarily support functionality. This research adapted the framework and used it to quantify network resilience.

The adapted framework takes a topological model¹⁰ that focuses on the design of the network topology, which differs from optimization model¹¹, data-driven model¹² and probabilistic model¹³. The topology model captures the nature of network-like infrastructures (such as the connection of different locations intrinsic to transportation systems), making it a popular choice for analyzing transportation systems^{10,14,15}. It offers an effective top-down approach to modeling interconnected infrastructure and understanding resilience despite the complexity. This research further exploits the capability of topological model in the resilience assessment from a perspective of safe-to-fail and resilience-by-design.

Introduction (line 113):

Overall, to fill the gaps, this research models a multi-modal public transportation network (MPTN) by integrating each mode of transportation network-by-network and compares the network resilience per state of being isolated and interconnected in each step of integration. Network resilience is quantified using a topology-based resilience framework, predicated on the "safe-to-fail" philosophy. This framework further leverages the potential of topological methods within the realm of resilience. The primary progression of our research is illustrated in Fig. 1. The remainder of the article unfolds with the Results section, which encompasses subsystem characteristics, sequential integration, robustness analysis, and interoperability analysis, followed by the Discussion section. Finally, the Methods section details the metrics used and the implementation of the null model.

(Big figure in the revised manuscript)

Figure 1 Schematic diagram of the research process. Comparison of resilience indicators and other characteristics metrics before and after network-by-network integration and per state of being isolated and interconnected lead to the conclusion. In particular, robustness and efficiency metrics are benchmarked with null model due to its sensitivity to network scale and changing parameters such as average node degree.

2 Besides this, some paragraphs should be moved. For example, lines 230-233 are suggested to be moved to the section where preparedness is discussed (lines 152-153, so it is clear why Gini values are used as preparedness indicator. Similarly, lines 246-249 are suggested to be moved to the section where robustness is discussed, and

Previous line 146-155 was the caption of table 1. To avoid confusion, we have moved it to table footnote. Besides, metrics such as Gini, robustness, and interoperability are now first introduced in the new Figure 1 and explained in the paragraph together with the resilience definition. Previous lines 230-233, 246-249, and 299-301 have been moved.

Following texts have been added to the manuscript.

lines 299-301 to be moved to the location where response and recovery measures are described.

Results > Characteristics of subsystems (line 179):

Gini coefficient is used to measure the extent to which sources of vulnerability are distributed across the network, signifying the existence of critical nodes that are not *safe to fail*. Network robustness is usually investigated through node percolation³² (or edge percolation³³), i.e., removing the nodes sequentially and observing the change of topological properties during network dismantling. We adopt the area under the degradation curve in the percolation as the robustness metric. Transportation interoperability is quantified by a relocation metric that measures the fraction of passengers in a disrupted station that can be relocated to nearby stations/stops, describing a post-disruption response pertaining to immediate recovery efforts (more details in Methods).

Table 1 Some characteristics and resilience-related metrics of the subsystems in this study

	MTR	FB	GMB	LR	FERRY	TRAM
$ V $	96	4279	4023	68	61	117
$ E $	204	9284	5689	156	94	124
$\langle k_{out} \rangle$	2.13	2.17	1.41	2.29	1.54	1.06
R	24	1443	715	22	38	7
S_0	1.00	≈ 1.00	0.79	1.00	0.15	1.00
l^{max}	27	83	107	26	7	84
$\langle l \rangle$	10.36	14.94	23.90	9.68	0.18	32.10
E	0.15	0.09	0.03	0.18	0.04	0.06
$E_{geospatial}$	0.71	0.73	0.29	0.84	0.05	0.53
$\langle l_e \rangle$	2020	1113	855	440	5882	245
$\sigma(l_e)$	2439	2626	1658	230	5342	86
Gini (ND)	0.174	0.348	0.252	0.147	0.215	0.051
Gini (BC)	0.490	0.611	0.757	0.456	0.727	0.188
r_b (Random)	0.207	0.191	0.070	0.213	0.071	0.040
r_b (ND-targeted)	0.080	0.075	0.021	0.090	0.040	0.024
r_b (BC-targeted)	0.076	0.074	0.023	0.107	0.041	0.027
RI ($d_{max} = 750$)	0.059	0.876	0.82	0.54	0.053	0.783
RI ($d_{max} = 1600$)	0.366	0.94	0.88	0.77	0.082	0.889

Note: $|V|$: number of stops; $|E|$: number of links; $\langle k_{out} \rangle$: average out-degree; R: number of directed routes (each for one direction); S_0 : relative size of largest strongly connected components ($S/|V|$), and $S_0 < 1$ means the network is not strongly connected; l^{max} : maximal shortest path length; $\langle l \rangle$: average shortest path length (path length between disconnected node pair is l^{max}); E : global efficiency; $E_{geospatial}$: geospatial modification of global efficiency (also known as detour index); $\langle l_e \rangle$: average edge lengths in haversine distance (in meter); $\sigma(l_e)$: standard deviation of edge lengths; Gini (ND) and Gini (BC): two variants of Gini coefficients based on the node degree (ND) and node betweenness centrality (BC), as the preparedness indicator; r_b : robustness indicator based on a certain attack strategy (random failure, ND-targeted, and BC-targeted); RI : global average relocation rate with certain relocation distance limit d_{max} , as the interoperability indicator.

3 Methods:
This paper takes edge length as the only factor when performing betweenness centrality analysis. This may be applicable to train and subway systems, as the speed of these forms of transportation may be comparable. Nevertheless, the speed of buses may vary due to the speed limit and traffic congestion. Thus, in the instance of a transportation network, the process of determining the shortest path, which is vital

This is an interesting question worth discussion. The speed and time modeling may benefit betweenness calculation, while it may also introduce bias to the outcomes and reduce its generalizability due to more assumptions on operational parameters (which can vary case by case). For the purposes of our research, current model is sufficient to support the conclusion on the topological impact of interconnectedness (focusing on the structure design). **Further explanations** as follows:

- Topology forms the backbone of the main structure, while edge weights might adjust the shortest paths.

	for centrality calculation, could be more precise if 'shortest time' was utilized over 'shortest distance'.	The key aspect here is our definition of node criticality, which is either "structural" or "operational" as derived from the betweenness metric. Our emphasis lies on the static and comparable nature of topology rather than the mutable weights. Besides, betweenness, as utilized in our research, merely supplements the degree metric in the targeted attack simulation and node homogeneity analysis. As such, we avoid deep modeling of shortest paths.  Literature has successfully used pure topological model to draw conclusion and offer useful insight into structure design of transport network (such as references R1-R5), while study incorporating weights tend to be more complicated due to case-by-case variation. The choice of weight, modeling approach, and real-world regional discrepancies can potentially introduce bias in the outcomes. Before the effect of weights are fully understood, we prefer to use an unweighted model as a prudent first step towards understanding topological implications on system resilience. We look forward to future research delving into this issue, possibly through comparative analysis of various edge-weighting methodologies. [R1] Berche, Bertrand, Christian Von Ferber, Taras Holovatch, and Yu Holovatch. "Resilience of public transport networks against attacks." The European Physical Journal B 71 (2009): 125-137. [R2] Duan, Yingying, and Feng Lu. "Structural robustness of city road networks based on community." Computers, Environment and Urban Systems 41 (2013): 75-87. [R3] Taylor, Michael. Vulnerability analysis for transportation networks. Elsevier, 2017. [R4] Wandelt, Sebastian, Xing Shi, and Xiaoqian Sun. "Estimation and improvement of transportation network robustness by exploiting communities." Reliability Engineering & System Safety 206 (2021): 107307. [R5] Reggiani, Aura, Peter Nijkamp, and Diego Lanzi. "Transport resilience and vulnerability: The role of connectivity." Transportation research part A: policy and practice 81 (2015): 4-15. Following texts have been added to the manuscript. Discussion (line 429): We look forward to future research delving into the effect of varying weighing methodologies and real-world discrepancies.
4	Is the actual route distance data available, so that the authors can replace the haversine distance with the actual route distance?	We employ the Haversine distance as a tool to identify interconnections among networks, evaluate walkability after a disruption, and depict the spatial restrictions intrinsic to a real-world network. As previously expressed in our response to the last comment, current estimation is sufficient to draw conclusion from a top-down perspective of spatial network topology design. Our analysis mainly involves making consequential comparisons of topological indicators. Distance measurements are kept consistent, acting as a relative rather than absolute scale.

		Furthermore, we have mentioned in the revision that developing weighted model and indicator variants will be a necessary step to real-world implementation, which bridges the theory and applications. Following texts have been added to the manuscript. Discussion (line 380): All the proposed resilience indicators can be adapted and improved for practical use, such as creating weighted variants, depending on the interest and requirement of stakeholders. Methods > Network efficiency (line 490): However, it is worth noting that further improvements could be made by incorporating data that provides actual travel distances, if necessitated by specific research questions and real-world applications.
5	Other comments:  1. This paper used different terms, such as 'intermodal' (line 21) and 'multimodal' in most other places. Unifying the terms is suggested to avoid any misunderstanding. 2. Line 58: 'is fail' => 'fails' 3. Line 137: Please specify that the number of nodes and edges are for the interconnected system. It isn't easy to understand from a first glance, as the prior sentence explains the subsystem. 4. Caption missing from Table 1. 5. I can understand that interoperability is an indicator of 'response'. But how it can represent 'recovery' is not clear since relocation does not restore the functionality of the system. 6. Line 181, should be Table 1? 7. Line 234, should be Table 2? 8. Line 275, should be Fig. 2? 9. Line 364, extra ')' at the end. In summary, I recommend major revision before this paper is accepted.	1) We thank the reviewer for the comments and have revised the texts. These two terms carry different implications in the context. We basically use "intermodal" to describe "edge", "transfer", and "connection" that are between two modes of transportation. "multimodal" is used for describing the "system" and "network" as a whole, denoting a holistic integration of multiple modes of transportation. To avoid any misunderstanding, we refrain from using the term "intermodal" prior to its initial definition in the manuscript. Following texts have been added to the manuscript. Results > Change in characteristics during one-by-one integration (line 225): Therefore, we analyzed the proximity to identify the interconnections and represented them with intermodal edges (edges connecting different modes of transportation in the graph model). 2-3) We appreciate it and have revised texts accordingly. 4) Line 146-155 was the caption of table 1. To avoid confusion, we have moved the metric descriptions to table footnote. Following texts in the manuscript have been moved. Table 2 Some characteristics and resilience-related metrics of the subsystems in this study

	MTR	FB	GMB	LR	FERRY	TRAM
$ V $	96	4279	4023	68	61	117
$ E $	204	9284	5689	156	94	124
$\langle k_{out} \rangle$	2.13	2.17	1.41	2.29	1.54	1.06
R	24	1443	715	22	38	7
S_0	1.00	≈ 1.00	0.79	1.00	0.15	1.00
l^{max}	27	83	107	26	7	84
$\langle l \rangle$	10.36	14.94	23.90	9.68	0.18	32.10
E	0.15	0.09	0.03	0.18	0.04	0.06
$E_{geospatial}$	0.71	0.73	0.29	0.84	0.05	0.53
$\langle l_e \rangle$	2020	1113	855	440	5882	245
$\sigma(l_e)$	2439	2626	1658	230	5342	86
Gini (ND)	0.174	0.348	0.252	0.147	0.215	0.051
Gini (BC)	0.490	0.611	0.757	0.456	0.727	0.188
r_b (Random)	0.207	0.191	0.070	0.213	0.071	0.040
r_b (ND-targeted)	0.080	0.075	0.021	0.090	0.040	0.024
r_b (BC-targeted)	0.076	0.074	0.023	0.107	0.041	0.027
RI ($d_{max} = 750$)	0.059	0.876	0.82	0.54	0.053	0.783
RI ($d_{max} = 1600$)	0.366	0.94	0.88	0.77	0.082	0.889

Note: $|V|$: number of stops; $|E|$: number of links; $\langle k_{out} \rangle$: average out-degree; R: number of directed routes (each for one direction); S_0 : relative size of largest strongly connected components ($S/|V|$), and $S_0 < 1$ means the network is not strongly connected; l^{max} : maximal shortest path length; $\langle l \rangle$: average shortest path length (path length between disconnected node pair is l^{max}); E: global efficiency; $E_{geospatial}$: geospatial modification of global efficiency (also known as detour index); $\langle l_e \rangle$: average edge lengths in haversine distance (in meter); $\sigma(l_e)$: standard deviation of edge lengths; Gini (ND) and Gini (BC): two variants of Gini coefficients based on the node degree (ND) and node betweenness centrality (BC), as the preparedness indicator; r_b : robustness indicator based on a certain attack strategy (random failure, ND-targeted, and BC-targeted); RI: global average relocation rate with certain relocation distance limit d_{max} , as the interoperability indicator.

5) We are not addressing the restoration of disrupted components. Our concept of "response" pertains to immediate recovery efforts. We have emphasized it in the following text. Furthermore, the question may also stem from varying interpretations of "functionality." Functionality can be understood from a people-centric resilience perspective, defined as the fraction of passenger that reach the destination despite disruptions. To this regard, response measures also contribute to a short-term increase in the functionality.

Following texts have been added to the manuscript.

Results > Characteristics of subsystems (line 184):

Transportation interoperability is quantified by a relocation metric that measures the fraction of passengers in a disrupted station that can be relocated to nearby stations/stops, describing a post-disruption response pertaining to immediate recovery efforts (more details in Methods).

Methods > Transportation interoperability (line 495):

Transportation interoperability is quantified by a metric of node relocation rate computed through the estimation of proximity and reachability (Eq. 4-5), which we understand as a short-term recovery (different from restoration of disrupted components).

6-9) We thank the reviewer for the comment and have corrected the typos.

REVIEWER COMMENTS #3

1	This paper describes results before the methods - very difficult to follow.	We have added brief methodology descriptions and a flowchart (new Figure 1) to the Introduction section to improve the readability. Please note that we keep the section order (introduction, results, discussion, methods) as required by the journal. Following texts have been added to the manuscript. Introduction (line 62 in the clean version): In previous studies, we have proposed a network resilience framework with three indicators upon the concept of "safe-to-fail", including preparedness, robustness, and interoperability^{8,9}. These indicators, respectively, are developed to examine whether the network topology is designed well to be ready to handle failure of any single component, to maintain critical functionality during progressive failures to a significant extent, and to interoperate with the remaining components to temporarily support functionality. This research adapted the framework and used it to quantify network resilience. The adapted framework takes a topological model¹⁰ that focuses on the design of the network topology, which differs from optimization model¹¹, data-driven model¹² and probabilistic model¹³. The topology model captures the nature of network-like infrastructures (such as the connection of different locations intrinsic to transportation systems), making it a popular choice for analyzing transportation systems^{10,14,15}. It offers an effective top-down approach to modeling interconnected infrastructure and understanding resilience despite the complexity. This research further exploits the capability of topological model in the resilience assessment from a perspective of safe-to-fail and resilience-by-design. Introduction (line 113): Overall, to fill the gaps, this research models a multi-modal public transportation network (MPTN) by integrating each mode of transportation network-by-network and compares the network resilience per state of being isolated and interconnected in each step of integration. Network resilience is quantified using a topology-based resilience framework, predicated on the "safe-to-fail" philosophy. This framework further leverages the potential of topological methods within the realm of resilience. The primary progression of our research is illustrated in Fig. 1. The remainder of the article unfolds with the Results section, which encompasses subsystem characteristics, sequential integration, robustness analysis, and interoperability analysis, followed by the Discussion section. Finally, the Methods section details the metrics used and the implementation of the null model. (Big figure in the revised manuscript)
----------	--	---

		 Figure 1 Schematic diagram of the research process. Comparison of resilience indicators and other characteristics metrics before and after network-by-network integration and per state of being isolated and interconnected lead to the conclusion. In particular, robustness and efficiency metrics are benchmarked with null model due to its sensitivity to network scale and changing parameters such as average node degree.
2	The concepts of "integration" (which seems to be much of the focus of the analysis) and "safe-to-fail" (which seems to be much of the focus of the title and introduction section) need to be better aligned.	We thank the reviewer for the suggestions. In essence, both two concept are integral to our research question. "Integration" is the step to examine the influence of interconnectedness on the resilience of the network, while the "safe-to-fail" is the foundational idea for developing resilience indicators, essentially determining how we measure resilience. The new Figure 1 shows the relationship between "safe-to-fail" and "integration". "Safe-to-fail" guides the metric design, and "integration" is process of connecting multiple modes to a single multimodal system for comparisons.  Figure 1 Schematic diagram of the research process. Comparison of resilience indicators and other characteristics metrics before and after network-by-network integration and per state of being isolated and interconnected lead to the conclusion. In particular, robustness and efficiency metrics are benchmarked with null model due to its sensitivity to network scale and changing parameters such as average node degree. Following texts have been added to the manuscript. Introduction (line 113): Overall, to fill the gaps, this research models a multi-modal public transportation network (MPTN) by integrating each mode of transportation network-by-network and compares the network resilience per state of being isolated and interconnected in each step of integration. Network resilience is quantified using a topology-based resilience framework, predicated on the "safe-to-fail" philosophy. This framework further leverages the potential of topological methods within the realm of resilience. The primary progression of our research is illustrated in Fig. 1.
3	Quite a lot of the introduction material is restatement of resilience literature found in many, many sources. It's recommended that the authors stick to what's new about this paper and how it fits in the landscape of resilience, not just providing a bunch of terms dealing with resilience.	We have removed the "restatement" content in the introduction section and included more texts discussing our approach. Following texts have been added to the manuscript. For the text that has been removed, kindly refer to the manuscript with tracked changes.

		Introduction (line 62): In previous studies, we have proposed a network resilience framework with three indicators upon the concept of "safe-to-fail", including preparedness, robustness, and interoperability^{8,9}. These indicators, respectively, are developed to examine whether the network topology is designed well to be ready to handle failure of any single component, to maintain critical functionality during progressive failures to a significant extent, and to interoperate with the remaining components to temporarily support functionality. This research adapted the framework and used it to quantify network resilience. The adapted framework takes a topological model¹⁰ that focuses on the design of the network topology, which differs from optimization model¹¹, data-driven model¹² and probabilistic model¹³. The topology model captures the nature of network-like infrastructures (such as the connection of different locations intrinsic to transportation systems), making it a popular choice for analyzing transportation systems^{10,14,15}. It offers a cost-effective top-down approach to modeling interconnected infrastructure and understanding resilience despite the complexity. This research further exploits the capability of topological model in the resilience assessment from a perspective of safe-to-fail and resilience-by-design. Introduction (line 113): Overall, to fill the gaps, this research models a multi-modal public transportation network (MPTN) by integrating each mode of transportation network-by-network and compares the network resilience per state of being isolated and interconnected in each step of integration. Network resilience is quantified using a topology-based resilience framework, predicated on the "safe-to-fail" philosophy. This framework further leverages the potential of topological methods within the realm of resilience. The primary progression of our research is illustrated in Fig. 1. The remainder of the article unfolds with the Results section, which encompasses subsystem characteristics, sequential integration, robustness analysis, and interoperability analysis, followed by the Discussion section. Finally, the Methods section details the metrics used and the implementation of the null model.
4	It is unclear how the Gini index measures network preparedness.	Following texts have been added to the manuscript. Results > Characteristics of subsystems (line 179): Gini coefficient is used to measure the extent to which sources of vulnerability are distributed across the network, signifying the existence of critical nodes that are not safe to fail. Methods > Network preparedness (line 438): From a topological perspective, structural vulnerability arises from node inequality. The presence of critical nodes contributes to the network's vulnerability, thus we regard these nodes as sources of vulnerability. The network is not prepared to handle failures related to these critical nodes if

		no additional protective measures are in place. In this research, network preparedness is gauged by analyzing how the vulnerability sources are distributed across the network from a homogeneity perspective. It indicates the extent to which the topology is designed to minimize the consequence of all possible node failures. In other words, every node in a prepared network is safe to fail itself. Gini coefficients of the node degree metric and node betweenness centrality metric are computed as the network preparedness indicator.
5	Both Mass Transit Railway and metro are abbreviated with MTR.	"Mass Transit Railway" is the name of the local metro company and its system. We have made a revision and consistently use MTR to represent the local metro.

REVIEWERS' COMMENTS

Reviewer #1 (Remarks to the Author):

The authors have addressed all my previous comments quite well and I would now recommend publication as is.

Reviewer #2 (Remarks to the Author):

I thank the authors for responding to my previous comments. I am good with the responses and corresponding revisions. This paper can be published in its current form.